

# Bioaccessibility of condensed tannins and their effect on the physico-chemical characteristics of lamb meat

Alejandro García Salas[1], Jose Ricardo Bárcena-Gama[2], Joel Ventura[1], Canuto Muñoz-García[3], José Carlos Escobar-España[4], Maria Magdalena Crosby[2] and David Hernandez[2]

[1] Department of Animal Production, Autonomous Agrarian University Antonio Narro, Saltillo, Coahuila, Mexico
[2] Livestock Program, Colegio de Postgraduados, Texcoco, Mexico State, México
[3] Faculty of Veterinary Medicine and Zootechnics No. 1, Autonomous University of Guerrero, Guerrero, Guerrero, Mexico
[4] Faculty of Agricultural Sciences, Campus IV, Autonomous University of Chiapas, Huehuetan, Chiapas, Mexico

Corresponding author
David Hernandez,
dave69455@hotmail.com

## ABSTRACT

The bioaccessibility of tannins as antioxidants in meat is essential to maximise their effectiveness in protecting the product. This property determines the amount of tannins available to interact with meat components, inhibiting lipid and protein oxidation and, consequently, prolonging shelf life and preserving the sensory quality of the product. The objective of this study was to evaluate the bioaccessibility of condensed tannins (CT) from *Acacia mearnsii* extract (AME) and their effect on the physico-chemical characteristics of fattened lamb meat. Thirty-six Dorset × Hampshire lambs (3 months old and $20.8 \pm 3.3$ kg live weight) were used. The lambs were distributed equally ($n = 9$) into four treatments: T1, T2, T3 and T4, which included a basal diet plus 0%, 0.25%, 0.5% and 0.75% of CT from AME, respectively. At the end of the fattening period, bioaccessibility was evaluated, the animals were slaughtered and a sample of the longissimus dorsi (LD) muscle was collected to assess colour, lipid oxidation, cooking weight loss and shear force on days 1, 4, 7 and 14 of shelf-life, in samples preserved at $-20\,°C$. In addition, the long chain fatty acid profile was analysed. A completely randomised design was used, and the means were compared with Tukey's test ($P < 0.05$). The mean lightness (L*), yellowness (b*) and hue (H*) values were higher for T3 and T4. The addition of CT did not affect ($P > 0.05$) redness (a*), cooking weight loss (CWL) or shear force (SF). T4 decreased ($P < 0.05$) stearic acid and increased cis-9 trans-12 conjugated linoleic acid (CLA). Bioaccessibility was higher in the supplemented groups (T1 < T2, T3 and T4). In conclusion, supplementing CT from AME in the diet of lambs did not reduce lipid oxidation, but T3 or T4 improved some aspects of meat colour and CLA deposition.

## INTRODUCTION

Oxidative processes are the main causes of meat quality deterioration and occur during the transition from muscle to meat, mainly during carcass processing or storage. The biochemical changes responsible for these processes disrupt the balance between the pro-oxidant and antioxidant systems *in vivo*, predisposing to premature oxidative reactions during the postmortem stage (*Cunha et al., 2018*). Lipid oxidation alters the pigmentation of the muscular haem group, transforming oxymyoglobin (red) to metmyoglobin (brown), affecting meat proteins and causing the development of secondary compounds that alter meat quality, such as changes in colour and texture and poor odour and flavour. This is due to the lack of endogenous antioxidants after slaughter (*Cunha et al., 2018*; *Gómez et al., 2018*). Changes in meat colour due to oxidation result in economic losses, as consumers choose meat based on its appearance (*Estrada-León et al., 2022*).

Antioxidants such as tertiary butylhydroquinone, butylated hydroxyanisole and butylated hydroxytoluene are widely used in commercial foods to prevent lipid oxidation and oxidative rancidity at the production, processing and storage stages (*Carocho, Morales & Ferreira, 2018*; *Cunha et al., 2018*).

The demand for foods of animal origin is increasing every day, and consumers demand products of excellent quality. This makes the use of natural antioxidants more acceptable than their synthetic counterparts, as they pose fewer risks to human health (*Zhao et al., 2018*). The addition of natural antioxidants such as condensed tannins (CT) to livestock diets helps delay lipid oxidation in meat and improves colour stability in meat from sheep (*Luciano et al., 2011*), goats (*Pimentel et al., 2021*), and cattle (*Gesteira et al., 2018*). They are also included to avoid biohydrogenation of fatty acids (FA) in the rumen, thus modifying the proportions of FA in the meat (*Gesteira et al., 2018*; *Frutos et al., 2020*). Supplementation of livestock diets with CT leads to variable responses, depending on their composition, concentration, dose and source. Additionally, it is important to consider the proportion of tannins that can be absorbed in the small intestine relative to the total amount of tannins consumed in the diet. In this regard, analysing bioaccessibility allows researchers to estimate the quantity of tannins available in the gut for transport into the cell (*Garrett, Failla & Sarama, 1999*). Thus, knowledge of bioaccessibility is necessary to accurately explain the bioactive effects of these metabolites in different tissues (*Palafox-Carlos, Ayala-Zavala & González-Aguilar, 2011*).

*Acacia mearnsii* extract (AME) has health benefits, including antioxidant, anticarcinogenic, anti inflammatory and antimicrobial activities (*Ogawa & Yazaki, 2018*). *Pimentel et al. (2021)* supplemented the diet of Boer goats with 0 to 48 g/kg AME as a source of CT, and observed better growth performance, increased yield of meat cuts, higher concentration of polyunsaturated fatty acids (PUFA) and better sensory characteristics of meat when the goats received 16 to 32 g of AME. In addition, supplementation with 20 g/kg of AME improved consumption, carcass yield, PUFA composition and meat texture in Santa Inés sheep (*Costa et al., 2021*). The objective of this study was to evaluate the bioaccessibility of CT and their effects on the physicochemical characteristics of meat from lambs supplemented with increasing levels of CT present in AME. It is hypothesised

that higher dietary tannin intake is associated with greater bioaccessibility and a better antioxidant effect in meat from lambs supplemented with CT from AME.

## MATERIALS AND METHODS

### Materials and reagents

AME, commercially marketed as SETA[®] (catalog # 89.717.268/0001-52, Estancia Velha –RS, Brazil) was used. Microfilters were obtained from Titan, catalog # 44513-NN (Mexico City, Mexico). Chromatography vial (catalog # 6ASV9-2P) and Whatman paper filters (no. 1) catalog # 1001 090) were purchased from Thermo Scientific (Waltham, MA, USA). The thiobarbituric acid reactive substances (TBARS) Assay Kit (catalog # 10009055), sodium methoxide (catalog # 156256), methanol (catalog # 494437), methanolic hydrochloric acid (catalog # 7647-01-0), ethanol (catalog # 51976), sodium carbonate (catalog # 71363), sodium sulphate (catalog # 7757-82-6) and standards of fatty acid methyl ester (FAME, catalog # 18919-1AMP) were purchased from Sigma-Aldrich (St. Louis, MI, USA). Trichloroacetic acid (catalog # 0414-04), hexane (catalog # 110-54-3), Folin reagent (catalog # 31360.264) and activated charcoal (catalog # 7440-44-0) were obtained from J.T. Baker[®] (Naucalpan, Mexico). Potassium carbonate (catalog # 104928) and gallic acid (catalog # 842649) were acquired from Merck[®] (Rahway, NJ, USA).

### Location

The study was carried out in the Ruminant Metabolic Unit of the Experimental Farm, and in the Animal Nutrition Laboratory of Postgraduate Program in Animal Husbandry of the Colegio de Postgraduados, Campus Montecillo, located in Texcoco, State of Mexico, Mexico.

### Animals

The animal performance test was conducted over a period of 85 days: 15 days of adaptation to the management conditions and consumption of the diets and 70 days of evaluation. Thirty-six F1 Dorset × Hampshire lambs with an initial live weight (LW) of 20.8 ± 3.3 kg were dewormed (Closantil oral[®] 5%, Lot # BBE048, one mL/5 kg LW, orally), vaccinated (Biobac[®] 11 Vias, Lot # B21005, 2.5 mL/animal, intramuscular route) and administered vitamins (Vigantol[®] ADE, Lot # KR26756, 0.5 mL/animal, intramuscular route). The animals were housed in individual raised metabolic cages (1.0 × 1.5 m) with access to a feed and water trough. The lambs were treated according to the 'Regulations for the use and care of animals destined for research at the Colegio de Postgraduados' (Animal Welfare Committee COBIAN/003/21). Feed was offered at a rate of 5% of the LW and provided twice daily (9:00 and 17:00 h), 60% in the morning and 40% in the afternoon, and water was offered *ad libitum*.

### Experimental diets

The concentration of CT in AME was 0.70 g/g of dry matter (DM). The experimental diets and their chemical composition are presented in Table 1. Four treatments were evaluated: T1, control diet; T2, T1 + 0.25% AME (CT = 1.75 g/kg DM); T3, T1 + 0.5% AME (CT

**Table 1 Experimental treatments (T) and chemical composition of the diets.**

| Items | Proportion in diet (%) | | | |
|---|---|---|---|---|
| | T1 | T2 | T3 | T4 |
| *Acacia mearnsii* extract (CT) | 0 | 0.25 | 0.5 | 0.75 |
| Corn ground | 65 | 65 | 65 | 65 |
| Soy bean meal | 8 | 8 | 8 | 8 |
| Oat hay | 20 | 20 | 20 | 20 |
| Molasses | 5 | 5 | 5 | 5 |
| Mineral premix[*] | 2 | 2 | 2 | 2 |
| **Chemical composition** | | | | |
| CT (g/kg DM) | 0.0 | 1.75 | 3.5 | 5.25 |
| Total phenols (g/kg DM) | 2.81 | 4.73 | 6.73 | 7.90 |
| Total proteins (%) | 13.41 | 13.41 | 13.41 | 13.41 |
| ME (Mcal/kg DM) | 2.81 | 2.81 | 2.81 | 2.81 |
| **Fatty acid profile (%)** | | | | |
| Myristic | 0.20 | 0.17 | 0.20 | 0.14 |
| Palmitic | 17.50 | 17.67 | 19.03 | 17.37 |
| Palmitoleic | 0.13 | 0.14 | 0.13 | 0.14 |
| Heptadecanoic | 0.08 | 0.08 | 0.09 | 0.08 |
| Stearic | 2.51 | 2.44 | 2.50 | 2.51 |
| Elaidic | 0.80 | 0.47 | 0.73 | 0.66 |
| Oleic | 30.87 | 32.68 | 32.81 | 33.18 |
| Linoleic | 45.06 | 44.13 | 42.46 | 43.03 |
| Arachidonic | 0.44 | 0.40 | 0.36 | 0.41 |
| Cis-11-Eicosenoic | 0.09 | 0.09 | 0.09 | 0.07 |
| Linolenic | 1.22 | 1.25 | 1.08 | 1.24 |

**Notes.**

Abbreviations: CT, Condensed tannins.

*Ca, 24%; Cl, 12%; Mg, 2%; P, 3%; K, 0.50%; Na, 8%; S, 0.50%; Cr, 5 mg/kg DM; Co, 60 mg/kg DM; I, 100 mg/kg DM; Fe, 2000 mg/kg DM; Mn, 4,000 mg/kg DM; Se, 30 mg/kg DM; Zn, 5,000 mg/kg DM; Lasalocid, 2,000 mg/kg DM; Vitamin A, 500 000 UI/kg; Vitamin D, 150,000 UI/kg; Vitamin E, 1,000 UI/kg.

= 3.5 g/kg DM); T4, T1 + 0.75% AME (CT = 5.25 g/kg DM). These percentages were adjusted based on the purity of the AME. The lambs were distributed randomly and equally ($N = 9$) among the four treatments. Each lamb represented the experimental unit and was identified by placing the treatment and replicate number on the front of each metabolic cage.

Samples of the experimental diets were collected every 15 days during the animal performance test, and composite samples were prepared at the end of the test for determination of dry matter (DM; method 934.01), total protein (TP; method 2001.11), ash (ASH; method 942.05) and ether extract (EE; method 920.39) (*Association of Official Analytical Chemists (AOAC), 2005*), as well as neutral detergent fibre (NDF) and acid detergent fibre (ADF) (*Van Soest, Robertson & Lewis, 1991*).

## Slaughter of animals

The 36 lambs were slaughtered after 85 days of fattening according to standard NOM-033-SAG/ZOO-2014, and in accordance with the regulations established by the Animal Welfare Committee (COBIAN/003/21) of the Colegio de Postgraduados. After 12 h of fasting and before transport to the slaughterhouse, the lambs were weighed on a digital scale (Torrey, CRS-HD, Monterrey, Mexico). At the time of slaughter, approximately 400 g samples of longissimus dorsi muscle were collected (*Guerrero, Ponce & Pérez, 2002*), placed in Ziploc® bags and stored at 4 °C for 24 h. Then, the samples were divided into four portions for evaluation at 1, 4, 7, and 14 days of shelf-life, placed back in Ziploc® bags and stored at −20 °C until analysis.

## Bioaccessibility of phenolic compounds

Feed intake was measured daily, and the mean dry matter intake (DMI) of each lamb was computed. To ascertain the bioaccessibility of phenolic compounds, a trial comprising a 5 day collection period was conducted from days 75 to 79 towards the end of the animal performance trial. For each lamb, the 24 h DMI was registered and feed samples (20 g) were collected. Likewise, faeces (10 g/lamb) were collected (09:00 h) in sterile tubes by rectal stimulation and using latex gloves and harness; the total amount of faeces per day was recorded. The feed and faeces samples were stored at −20 °C and protected from the light until analysis of phenolic compounds as indicated below. Bioaccessibility was calculated as the difference between the total phenolic content consumed in the diet minus the total phenolic content in the faeces, and expressed as a percentage.

## Determination of the total phenolic content

To determine the total phenolic content, three 50 mg samples of the diet, faeces and meat samples from each treatment were weighed. The samples were placed in 15 mL plastic tubes, and 10 mL of 80% ethanol was added. The tubes were capped and incubated in an ultrasonic bath for 10 min. The bath was switched off for 5 min and then subsequently switched on for another 10 min. Then, the tubes were centrifuged at 5,000 g for 10 min at 4 °C. The supernatant was refrigerated until analysis (*Makkar, 2003*).

The Folin-Ciocalteu method was used to determine the total phenolic content. The following were added to a glass test tube measuring 100× 120 mm: 500 μL of the cold extract, 25 μL of a mixture of Folin reagent 1N and water (1:1, v/v) and 975 μL of 2.5% sodium carbonate. After 60 min, the absorbance was read at 740 nm in a spectrophotometer (model 336008; Thermo Fisher Scientific, Waltham, MA, USA). A fresh solution of gallic acid at a concentration of 0.02 mg/mL was used to construct a standard curve. The total phenolic content is expressed as mg/g DM.

## Physico-chemical properties
### Colour

The preserved meat samples were removed from the Ziploc® bag and incubated at room temperature (20 °C) for 30 min to thaw them. The colour readings of the preserved samples were taken after 1, 4, 7, and 14 days of storage. A Minolta CR-400 colorimeter (Konica Minolta Sensing, Inc., Tokyo, Japan) was used with measurements in the CIELAB space

at a diameter of eight mm, a D65 illuminant and a viewing angle of 0°. Three colour measurements were taken at different points on the samples. The colour measurements are expressed as the CIE L* (lightness), a* (redness) and b* (yellowness) values. The hue angle (H*), which defines the colour, was calculated as arctan-1 (b*/a*) expressed in degrees, and chroma (C*) was calculated as (a*2+b*2)0.5 (*Hernández et al., 2016*).

### Lipid oxidation

Lipid oxidation was assessed after 1, 7, and 14 days of storage by measuring 2-thiobarbituric acid reactive substances with the TBARS Assay Kit, according to the method described by *Siu & Draper (1978)*. Meat samples (2.5 g) were homogenised with 12.5 mL of distilled water; then, 12.5 mL of 10% trichloroacetic acid was added to precipitate proteins. The mixture was vortexed (Labnet, Edison, NJ, USA). The homogenate was centrifuged at 10,000 g for 10 min at 4 °C. The homogenised solution was filtered through Whatman No. 1 filter paper; four mL of the filtrate was taken, placed in a Pyrex glass tube and mixed with one mL of 0.06 M thiobarbituric acid. The samples were incubated in a water bath (PolyScience®, IL, USA) at 80 °C for 90 min. Finally, the absorbance was read at 532 nm with a Spectrophotometer (CARY 1-E, Varian, ON, Canada). The assay was calibrated with a solution of a known concentration of 1,1,1,3,3-tetraethoxypropane (TEP). The results are expressed as mg of malondialdehyde (MDA) per kg of meat.

### Cooking weight loss

The cooking weight loss (CWL) was determined by using the method described by *Honikel (1998)* on meat samples stored for 1, 4, 7, and 14 days. The meat samples (80 g) were placed in polyethylene bags, which were then placed in a water bath (PolyScience, IL, USA) until the internal temperature of the sample reached 70 °C. The cooked samples were allowed to cool for 30 min and dried until they reached 20−25 °C. The CWL was calculated as the difference in weight before and after cooking.

### Shear force

After measuring the CWL, the same samples were used to determine the shear force (SF). The meat samples were cut parallel to the axis of the muscle fibres to a size of 1 cm high ×1 cm wide ×2 cm long. These pieces were cut perpendicular to the fibres, using a Warner-Bratzler cutting blade. For each sample, the maximum SF was recorded using a Texturometer model TAXT2 (Stable Microsystems Corp., Vienna, UK). The value reported (kgF/cm$^2$) for each sample is the mean of three cuts (*Gómez et al., 2018*).

### FA profile

The modified esterification technique of *Sukhija & Palmquist (1988)*, *Palmquist & Jenkins (2003)* and *Jenkins (2010)* was used to determine the FA profile in the feed and meat samples. Each sample (0.5 g) was placed in a 16 mL polypropylene tube with 2 mL of sodium methoxide in methanol 0.5 M, then vortexed (Labnet International, Inc., Edison, NJ, USA), incubated in a water bath at 50 °C for 10 min and cooled at room temperature (20 °C) for 5 min. Next, 3 mL of methanolic hydrochloric acid (5%, 1.37 M) was added. The mixture was vortexed and incubated in a water bath at 78 °C for 12 min. Then, it was

cooled to room temperature (20 °C) for 7 min, and 3 mL of hexane and 5 mL of potassium carbonate (6%, 0.43 M) were added. The mixture was vortexed and then centrifuged at 2,500 g for 5 min in a refrigerated centrifuge (Beckman J2-HS, GMI, Ramsey, MN, USA). The supernatant was transferred to a polypropylene tube (16 mL) containing 0.5 g of sodium sulphate and 0.1 g of activated charcoal. The sample was vortexed and then centrifuged at 1,500 g for 5 min at 4 °C. The hexane phase was extracted with microfilters (17 mm–0.45 $\mu$m) and transferred to a chromatography vial. The sample was analysed in a gas chromatograph (HP 6890, Agilent, Santa Clara, CA, USA) with an automatic injector (HP 7683, Agilent), an autosampler tray and a SP$^{\circledR}$ 2,560 capillary column 100 m × 0.25 mm × 0.2 $\mu$m (film; Supelco, Bellefonte, PA, USA) at 29 psi. The detector conditions were: air flow of 330 mL/min and hydrogen flow of 33 mL/min at 260 °C. The injector conditions (250 °C) were: helium as a carrier gas and an auxiliary flow of 18 mL/min at 29 psi for 62 min. The temperature ramps were: ramp 1, 1 °C/min, 140 °C for 2.95 min; ramp 2, 3 °C/min, 210 °C; and ramp 3, 0.7 °C/min, 235 °C. Fatty acid methyl ester (FAME) Mix C4-C24 from Supelco (49905-U; Bellefonte, PA, USA), trans-11-vaccenic methyl ester (18919-1AMP; Supelco, Bellefonte, PA, USA), and conjugated linoleic acid > 90% were used as standards to determine the retention times of FAMEs.

### Data analysis
The data were analysed by using a completely randomised design with four treatments and nine replicates per treatment. The variables were analysed with PROC MIXED models. The FA results were processed using PROC GLM (*SAS, 2002*) (version 9.4, Cary, NC, USA). The means were compared with Tukey's test ($P < 0.05$).

## RESULTS AND DISCUSSION
### Bioaccessibility of phenolic compounds
The total phenolic content in the studied diets increased in direct proportion to the amount of CT supplementation ($P < 0.05$; Table 2). The total phenolic content in the faeces was similar ($P > 0.05$) between T1 and T4, although T4 had 236% more of these metabolites in the diet. Supplementation with 0.25% and 0.5% AME was associated with the lowest ($P < 0.05$) excretion of phenolic compounds in faeces. The bioaccessibility and the amount of phenolic compounds deposited in the meat were higher in the supplemented groups ($P < 0.05$) compared with the control group. However, the proportion of phenolic compounds present in meat was <10% (4.21%, 9.9%, 7.1% and 6.2% for T1, T2, T3 and T4, respectively), with respect to the initial concentration in the diet. There was no difference ($P > 0.05$) among the supplemented groups regarding the phenolic compounds deposited in the meat.

CT are polymers of flavonol units (flavan-3-ol) linked by carbon–carbon bonds that are not susceptible to anaerobic enzymatic degradation in the rumen. They can be oxidatively degraded in acidic environments to anthocyanidins and are therefore known as proanthocyanidins (*Waghorn, 2008*). Bioactive compounds such as tannins must be released from the feed matrix to exert their biological effect. This process is called phenolic bioaccessibility; it is the fraction of polyphenols available for intestinal absorption. Phenolic

**Table 2  Bioaccessibility of total phenols in lambs supplemented with condensed tannins from *Acacia mearnsii* extract (AME), at different shelf-life times.**

| | Treatment[¥] | | | | |
|---|---|---|---|---|---|
| | T1 | T2 | T3 | T4 | SEM |
| Total phenols | —————————(mg/g DM)————————- | | | | |
| Feed | 2.61[d] | 4.54[c] | 6.52[b] | 8.57[a] | 0.498 |
| Faeces | 1.24[a] | 0.91[c] | 1.02[bc] | 1.24[a] | 0.039 |
| Meat | 0.11[b] | 0.47[a] | 0.47[a] | 0.54[a] | 0.028 |
| Bioaccessibility (%) | 51.85[b] | 79.78[a] | 84.32[a] | 84.92[a] | 3.637 |

Notes.

Abbreviations: DM, Dry matter.

[¥]Treatments included a basal diet plus 0, 0.25, 0.5 and 0.75% of AME for T1, T2, T3 and T4.

[abcd]Means with different letters in each row indicate significant differences ($P < 0.05$).

bioavailability is the fraction of polyphenols released into the circulation for metabolism in tissues (*Karás et al., 2017*). The bioaccessibility of phenolic compounds determined in the present study ranged from 79.7% to 84.9% in the supplemented groups. Although high, it is estimated that only 5% to 10% of ingested CT is absorbed intact in the small intestine (*Bell et al., 2000*), suggesting low bioavailability of these metabolites. The bioaccessibility of CT is related to deglycosylation catalysed by lactase-floricin hydrolase, which is present in small intestinal epithelial cells (*Gonzales et al., 2015*). On the other hand, phenolic compounds are transported in epithelial cells by sodium-glucose transport proteins located in the small intestinal mucosa (*Day et al., 2000*). Within enterocytes, phenolic compounds undergo a series of methylation, glucuronidation and sulphation reactions to attenuate their prooxidant and cytotoxic effects (*Donovan et al., 2006*). It is unclear whether elevated phenolic concentrations compete with nutrients for small intestinal absorption sites and impede their bioavailability. Based on CT intake, the percentages of excreted phenolic compounds were 20.5%, 15.6% and 13.1% for T2, T3, and T4, respectively, indicating minimal losses of these metabolites. The low levels of CT detected in the meat (9.9%, 7.1% and 6.2% for T2, T3 and T4 respectively) suggest limited bioavailability, potentially due to degradation or elimination in the urine. It may also be explained by the instability of phenolic compounds with respect to light, oxygen, pH and temperature during storage (*Cunha et al., 2018*). These factors can be related to the minimal effects that AME exerted on the physicochemical characteristics of the lamb meat. In support of this view, *Borges et al. (2018)* isolated CT monomers, with 82% found in plasma and urine, 12.3% excreted in faecal matter and 5.2% present in tissues, consistent with the levels of CT detected in the meat in the present study.

## Colour

Supplementation with 0.5% and 0.75% AME resulted in the highest ($P < 0.05$; Table 3) mean L* in the lamb meat. There was an interaction ($P < 0.02$) between the treatment and storage time, with higher L* observed on day 14 compared with day 1 for T3 and T4. a* was not different among the treatments ($P > 0.05$); however, there was a negative linear effect ($P < 0.05$) over time, with the lowest value ($P < 0.05$) on day 14. b* was highest ($P < 0.05$) for T3 and T4; there was a positive linear effect ($P < 0.05$) between the treatment and
**Table 3  Meat colour of lambs supplemented with condensed tannins from *Acacia mearnsii* extract (AME), at different shelf-life times.**

| | Days | Treatment[¥] T1 | T2 | T3 | T4 | x̄Days | SEM | P-value Treat | Day | Treat*Day | Linear | Quadratic |
|---|---|---|---|---|---|---|---|---|---|---|---|---|
| | 1 | 36.81[b] | 37.79[ab] | 39.87[ab] | 42.17[a] | 39.2 ± 0.51[c] | 0.78 | ** | ** | 0.0281 | ** | 0.867 |
| | 4 | 37.97[b] | 39.62[ab] | 42.67[a] | 42.27[a] | 41.1 ± 0.43[b] | | | | | | |
| L* | 7 | 39.05[b] | 39.80[b] | 42.30[ab] | 43.96[a] | 41.3 ± 0.38[b] | | | | | | |
| | 14 | 39.63[bc] | 40.88[bc] | 45.30[ab] | 45.42[a] | 42.8 ± 0.45[a] | | | | | | |
| | x̄ | 38.36[b] | 39.52[b] | 42.53[a] | 43.95[a] | | | | | | | |
| | 1 | 17.42[b] | 17.90[ab] | 19.91[a] | 19.55[ab] | 18.7 ± 0.22[a] | 0.35 | 0.14 | ** | ** | 0.07 | 0.511 |
| | 4 | 19.00 | 19.34 | 19.02 | 18.85 | 19.0 ± 0.25[a] | | | | | | |
| a* | 7 | 19.04 | 18.74 | 18.50 | 19.06 | 18.8 ± 0.17[a] | | | | | | |
| | 14 | 17.63[ab] | 16.78[bc] | 11.57[d] | 13.42[cd] | 14.8 ± 0.38[b] | | | | | | |
| | x̄ | 18.27 | 18.2 | 17.25 | 17.64 | | | | | | | |
| | 1 | 4.63 | 4.50 | 5.01 | 6.06 | 5.0 ± 0.24[d] | 0.28 | ** | ** | ** | ** | 0.682 |
| | 4 | 5.49 | 5.39 | 6.03 | 7.06 | 6.0 ± 0.18[c] | | | | | | |
| b* | 7 | 6.62 | 6.33 | 7.44 | 7.87 | 7.1 ± 0.20[b] | | | | | | |
| | 14 | 6.52[bc] | 7.84[b] | 11.26[a] | 10.46[a] | 9.0 ± 0.23[a] | | | | | | |
| | x̄ | 5.81[b] | 6.01[b] | 7.43[a] | 7.86[a] | | | | | | | |
| | 1 | 23.12[ab] | 18.48[bc] | 24.40[a] | 26.16[a] | 23.0 ± 0.50[a] | 0.33 | 0.01 | ** | ** | 0.102 | 0.002 |
| | 4 | 24.86[a] | 24.90[a] | 19.97[b] | 19.44[b] | 22.4 ± 0.25[a] | | | | | | |
| C* | 7 | 20.16[b] | 19.81[b] | 19.97[b] | 20.67[b] | 20.1 ± 0.20[b] | | | | | | |
| | 14 | 18.86 | 18.60 | 16.40 | 17.26 | 17.8 ± 0.25[c] | | | | | | |
| | x̄ | 21.75[a] | 20.44[b] | 20.18[b] | 21.0[ab] | | | | | | | |
| | 1 | 14.85 | 14.01 | 14.02 | 16.97 | 15.0 ± 0.62[d] | 1.00 | ** | ** | ** | ** | 0.795 |
| | 4 | 16.04 | 15.51 | 16.64 | 21.00 | 17.5 ± 0.58[c] | | | | | | |
| H* | 7 | 18.97[a] | 18.64[a] | 21.91[a] | 22.36[b] | 20.5 ± 0.49[b] | | | | | | |
| | 14 | 20.35[b] | 25.10[b] | 44.67[a] | 38.88[a] | 32.2 ± 1.23[a] | | | | | | |
| | x̄ | 17.55[b] | 18.31[b] | 24.56[a] | 24.80[a] | | | | | | | |

**Notes.**

Abbreviations: Treat, treatment; L*, luminosity; a*, Red index; b*, Yellow index; C*, Chromaticity; H*, Tonality.

[¥]Treatments included a basal diet plus 0, 0.25, 0.5 and 0.75% of AME for T1, T2, T3 and T4.

[abcd]Means with different letters in each row, and the interaction of rows with columns, indicate significant differences.

*$P < 0.05$.

**$P < 0.0001$.

storage time. T1 had a higher mean C* ($P < 0.05$) than T2 and T3, but it was similar to T4. The inclusion of CT in the diet had a negative linear effect ($P < 0.05$) on C*. On day 14, T3 and T4 had a high H* ($P < 0.05$). This variable increased linearly during storage ($P < 0.05$).

The L*, a*, b*, C* and H* values obtained in this study are similar to those reported in the literature for sheep meat (*Callejas, 2016*; *Grochowska et al., 2019*; *Navarro & Rodríguez-González, 2022*). AME supplementation had the most pronounced effect on colour, denoted by brighter meat when the diets contained 0.5% and 0.75% AME. This is a positive attribute as L* is the variable most closely associated with the visual appearance of meat, making it more attractive to consumers (*Salinas et al., 2020*; *Carrillo-Lopez et al., 2021*). In addition, it has been suggested that a bright cherry red colour is perceived by consumers as an

indicator of quality (*Ngapo, Brana Varela & Rubio Lozano, 2017*). In the present study, a*
did not show a clear response with CT supplementation, but generally decreased with
storage, as did C*. On the other hand, L*, b* and H* increased over time. *Navarro &
Rodríguez-González (2022)* observed similar results for a*, b* and L*. As the meat matures,
the myofibrils gradually separate. This allows more light to penetrate the meat. In addition,
this process facilitates the interaction of oxygen with myoglobin, transforming it into
oxymyoglobin and imparting a redder colour (*Callejas, 2016*). Tannin supplementation
counteracts meat colour degradation during storage due to their antioxidant activity
and interaction with myoglobin (*Luciano et al., 2011*; *Gómez et al., 2018*). In this study,
although T3 and T4 produced whiter meat, CT supplementation did not alter the redness
colour of the meat. Although a* was stable between days 1 and 7 of storage (18.7 to 18.8)
in all treatments, the lowest mean value was recorded on day 14 (14.8), regardless of the
treatment. In addition, the lack of change in a* with CT supplementation could be related
to the age of the lambs, as cherry red shades associated with fresh meat have been found in
young animals, such as the lambs used in the present study. In adult animals, the amount
of pigment and a* increase, while brightness decreases (*Radzik-Rant et al., 2020*). Values
higher than 34 (L*) and 9.5 (a*) are acceptable to consumers of fresh lamb meat (*Suliman
et al., 2021*). *Callejas (2016)* mentioned that values of ≥42 (L*) and ≥7 (a*) are associated
with higher product acceptability. The a* values observed in the present study exceed
those reported by these authors. However, L* increased and better quality attributes of
meat were perceived when lambs were supplemented with 0.5% and 0.75% AME because
of antioxidant effect of CT (*Zhong et al., 2015*). Several factors influence meat colour,
including genotype, diet, age (*Radzik-Rant et al., 2020*), storage temperature, the type of
packaging and muscle biochemistry, among others (*King, Shackelford & Wheeler, 2011*).
Understanding the interaction of CT with these factors could explain the response in meat
colour when these bioactive compounds are added to the diet.

## Lipid oxidation

The inclusion of 0.5% and 0.75% AME resulted in a higher mean ($P < 0.05$; Table 4) MDA
concentration in lamb meat, which is associated with greater lipid oxidation. In addition,
the MDA levels increased over time ($P < 0.05$) in all treatments, with the highest levels
on day 14. Oxidative processes cause meat deterioration, a phenomenon related to the
content of easily oxidisable substrates and antioxidants (*Bekhit et al., 2013*). Approximately
half of the FA in lamb meat are unsaturated fatty acids (UFA) (*Hajji et al., 2016*). The
high proportion of UFA is conducive to meat oxidation (*Kumar et al., 2015*). Previous
studies have reported positive effects on the oxidative stability of meat when tannins were
added to the diet of ruminants (*Luciano et al., 2011*; *Luciano et al., 2019*). Another study
documented a reduction in lipid oxidation denoted by a decrease in the TBARS values by
78% and 62% in ground lamb meat with the addition of 3% *Rhus coriaria* and *Berberis
vulgaris* extracts, respectively, after 9 days of storage (*Aliakbarlu & Mohammadi, 2015*).
However, such effects were not observed in the current study. Our results are consistent
with the study by *Valenti et al. (2019)*. Those authors evaluated different types and sources
of tannins and found no effect on the oxidative stability of lamb meat supplemented with

**Table 4  Lipid oxidation in meat from lambs supplemented with condensed tannins from *Acacia mearnsii* extract (AME), at different shelf-life times.**

|  | | Treatment[¥] | | | | | | P- value | | | | |
|---|---|---|---|---|---|---|---|---|---|---|---|---|
|  | Days | T1 | T2 | T3 | T4 | $\bar{x}$Day | EEM | Treat | Day | Treat*Day | Linear | Quadratic |
| TBARS | 1 | 0.15[b] | 0.10[c] | 0.56[ab] | 0.53[ab] | 0.33[a] | 0.147 | 0.049 | 0.002 | 0.218 | 0.019 | 0.895 |
|  | 7 | 0.16[bc] | 0.12[bc] | 0.79[a] | 0.57[ab] | 0.41[a] |  |  |  |  |  |  |
|  | 14 | 0.49 | 0.45 | 1.46 | 1.45 | 0.96[b] |  |  |  |  |  |  |
|  | $\bar{x}$ Trat | 0.27[b] | 0.22[b] | 0.93[a] | 0.85[a] |  |  |  |  |  |  |  |

Notes.

Abbreviations: Treat, treatment.

[¥]Treatments included a basal diet plus 0, 0.25, 0.5 and 0.75% of AME for T1, T2, T3 and T4. TBARS[1], milligrams of malondialdehyde per kilogram of meat (mg MDA kg$^{-1}$).

[abcd]Means with different letters in each row, and the interaction of rows with columns, indicate significant differences ($P < 0.05$).

4% AME (condensed tannins), *Castanea sativa* extract (hydrolysable ellagitannins) and *Cesalpinia spinosa* extract (hydrolysable gallotannins).

The discrepancies in the results may also be related to the type of tannins, the dose used and the interaction with the diet, factors that do not allow clear conclusions to be drawn regarding the effects of tannins on the oxidative stability of meat (*Luciano et al., 2019*). Regardless of the treatment, the TBARS levels increased with the storage time, consistent with the literature (*Luciano et al., 2011*). However, it is important to note that the TBARS levels were < 2 mg MDA/kg, a value above which off-flavours and rancidity of the meat are perceived (*Howes et al., 2015*).

## CWL and SF

The CWL did not vary among the treatments ($P > 0.05$; Table 5). However, there was a linear effect ($P < 0.05$) on water loss over time, with greater loss during the first 4 days. The CWL recorded in this study is higher than what *Gómez et al. (2018)* observed; they provided CT in the diet of lambs and reported a CWL of 18.9% to 24.1%. However, our CWL is similar to the data reported by *Estrada-León et al. (2022)* on lamb meat (32.7% to 33.6%).

The water-holding capacity (WHC), as measured by CWL, is associated with greater shrinkage and stiffness of meat myofibrillar structures, where much of the water is retained. It is also affected by temperature, as proteins are denatured at 45 to 80 °C, releasing more water, but these losses gradually decrease above 80 °C (*Hughes et al., 2014*). Oxidation of meat reduces the WHC between muscle myofibrils, a phenomenon that increases the loss of muscle fluid (*Huff-Lonergan & Lonergan, 2005*). In the present study, the CWL did not change despite the high levels of oxidation observed for T3 and T4. The use of antioxidants could improve the WHC by maintaining membrane integrity and protein cross-linking (*Estévez, 2011*). Similarly, the inclusion of CT in the diet reduces CWL and is related to the amount of fat in the meat. CT preserves lipids, and the CWL corresponds to the loss of water and fat, especially in meat rich in UFA with low melting points (*Carvalho et al., 2014*). Consistently, *Gómez et al. (2018)* reported a decrease in the CWL as the dietary tannin content increased. In contrast, the results of the present study indicated greater lipid oxidation when high CT levels were added to the lamb diet (T3 and T4), which could explain the increase in the CWL for T4, although this change was not significant. Regardless

**Table 5 Cooking weight loss (CWL) and shear force (SF) in meat from lambs supplemented with condensed tannins from *Acacia mearnsii* extract (AME), at different shelf-life times.**

| | | Treatment[¥] | | | | | | P-value | | | | |
|---|---|---|---|---|---|---|---|---|---|---|---|---|
| | | T1 | T2 | T3 | T4 | $\bar{x}$Days | SEM | Treat | Day | Treat*Day | Linear | Quadratic |
| CLW (%) | 1 | 33.14 | 35.19 | 33.40 | 34.33 | $34.0 \pm 0.41^{b}$ | 0.82 | 0.056 | ** | 0.058 | 0.032 | 0.200 |
| | 4 | 31.60 | 34.76 | 32.74 | 36.30 | $33.8 \pm 0.41^{b}$ | | | | | | |
| | 7 | 30.02 | 29.56 | 29.67 | 33.48 | $30.7 \pm 0.48^{a}$ | | | | | | |
| | 14 | 29.95 | 27.87 | 29.27 | 32.14 | $29.8 \pm 0.97^{a}$ | | | | | | |
| | $\bar{x}$ | 31.17 | 31.85 | 31.27 | 34.06 | | | | | | | |
| SF (kgF/cm$^2$) | 1 | 3.21 | 2.88 | 2.39 | 2.38 | $2.72 \pm 0.11^{a}$ | 0.18 | 0.702 | ** | 0.0006 | 0.296 | 0.607 |
| | 4 | 2.83 | 2.41 | 2.35 | 2.33 | $2.48 \pm 0.13^{ab}$ | | | | | | |
| | 7 | 2.22 | 2.14 | 2.24 | 2.38 | $2.25 \pm 0.14^{b}$ | | | | | | |
| | 14 | 1.26 | 1.26 | 1.54 | 1.35 | $1.35 \pm 0.08^{c}$ | | | | | | |
| | $\bar{x}$ | 2.38 | 2.17 | 2.13 | 2.11 | | | | | | | |

**Notes.**

Abbreviations: Treat, treatment.

[¥]Treatments included a basal diet plus 0, 0.25, 0.5 and 0.75% of AME for T1, T2, T3 and T4.

[abcd]Means with different letters in columns indicate significant differences ($P < 0.05$).

[**]$P < 0.0001$.

of the treatment, the greatest weight loss was observed during the first 4 days of storage, and the weight loss in later days could be explained by water loss through dripping and evaporation, denaturation of sarcoplasmic and cytoskeletal proteins, and by the release of water from the intracellular and extracellular compartments during meat storage (*Giráldez et al., 2021*).

SF showed no change ($P > 0.05$; Table 5) among the treatments; however, there was a linear effect ($P < 0.05$) during storage, with a lower value ($P < 0.05$) on day 14. In agreement with our results, *Pimentel et al. (2021)* observed no differences in the SF in cooked goat meat after supplementation with 0, 16, 32 and 48 g AME/kg DM. Similarly, there were no changes in the texture of lamb meat when grape pomace was provided as a source of CT, with SF values ranging from 5.38 to 6.85 (*Gómez et al., 2018*). These values being higher than those recorded in the current study. Moreover, *Hernández et al. (2023)* reported SF values similar to the present study, with no differences between treatments when 1.5% or 2.5% CT was added to the diet of lambs. Texture is the most important palatability characteristic in lamb meat and influences consumer acceptance (*Giráldez et al., 2021*). Oxidation of myofibrillar proteins leads to hardening of the meat (*Estévez, 2011*). The collagen content in intramuscular connective tissue is also negatively correlated with meat texture (*Zhao et al., 2018*). CT supplementation protects endogenous proteases from oxidation and results in lower SF during the meat maturation process (*Morán et al., 2012*). Although there were no differences in the SF among the treatments, the mean SF in the present study ranged from 2.11 to 2.38, which is lower than the value classified as medium tenderness (4.5 kgF/cm$^2$) by *Gómez et al. (2018)*, indicating more tender meat.

### FA composition of meat

The inclusion of 0.75% AME in the lamb diet was associated with a low ($P < 0.05$; Table 6) stearic acid and a high cis-9-trans-12 CLA acid meat content, without affecting ($P > 0.05$)

**Table 6  Fatty acid composition (% of total methylated fatty acid esters) in meat from lambs supplemented with condensed tannins from *Acacia mearnsii* extract (AME), at different shelf-life times.**

| | Treatment[¥] | | | | SEM |
|---|---|---|---|---|---|
| | T1 | T2 | T3 | T4 | |
| Myristic | 2.42 | 2.36 | 2.48 | 2.54 | 0.16 |
| Pentadecanoic | 0.34 | 0.35 | 0.33 | 0.39 | 0.05 |
| Palmitic | 24.54 | 24.65 | 24.82 | 25.11 | 0.74 |
| Heptadecanoic | 1.33 | 1.56 | 1.52 | 1.54 | 0.10 |
| Stearic | 15.55[a] | 15.72[a] | 14.80[ab] | 13.54[b] | 0.49 |
| **Total SFA** | 44.20 | 44.66 | 43.69 | 43.14 | 0.85 |
| Palmitoleic | 2.30 | 2.29 | 2.26 | 2.25 | 0.09 |
| Cis-10-heptadecanoic | 0.72 | 0.88 | 0.94 | 0.84 | 0.07 |
| Helladic | 3.14 | 3.43 | 3.92 | 3.74 | 0.61 |
| Oleic | 41.51 | 40.83 | 41.39 | 40.73 | 1.06 |
| **Total MUFA** | 47.69 | 47.44 | 48.51 | 47.57 | 0.63 |
| Linoleic | 4.76[ab] | 4.88[ab] | 4.61[b] | 6.02[a] | 0.36 |
| Cis-11-Eicosanoic | 0.05 | 0.05 | 0.06 | 0.05 | 0.01 |
| Linolenic | 0.07 | 0.07 | 0.09 | 0.09 | 0.01 |
| Cis-9-trans-11CLA | 0.10 | 0.09 | 0.13 | 0.12 | 0.01 |
| Trans-10-Cis—12 CLA | 0.00[b] | 0.00[b] | 0.00[b] | 0.02[a] | 0.001 |
| Arachidonic | 1.46 | 1.30 | 1.07 | 1.28 | 0.16 |
| **Total PUFA** | 6.46 | 6.33 | 5.98 | 7.61 | 0.49 |
| **MUFA/SFA** | 1.08 | 1.06 | 1.10 | 1.10 | 0.03 |
| **PUFA/SFA** | 0.14 | 0.14 | 0.13 | 0.17 | 0.01 |

Notes.

Abbreviations: SFA, saturated fatty acids; MUFA, monounsaturated fatty acids; PUFA, polyunsaturated fatty acid.

[¥]Treatments included a basal diet plus 0, 0.25, 0.5 and 0.75% of AME for T1, T2, T3 and T4.

[ab]Means with different letters in each row indicate significant differences ($P < 0.05$).

the other FA. In a similar study, the addition of 0, 10, 30 and 50 g of AME/kg DM in the diet decreased saturated fatty acids (SFA) and increased UFA in beef (*Gesteira et al., 2018*), findings similar to those observed in this study with stearic acid and cis-9 trans-12 CLA. In contrast, *Gómez et al. (2018)* did not report any significant changes in the FA profile deposited in the meat of lactating lambs when their mothers were supplemented with 5% and 10% of grape pomace as a source of CT.

High lipid and SFA levels in red meat increase plasma cholesterol and the incidence of cardiovascular disease and atherosclerosis (*Parodi, 2016*). For this reason, reducing the consumption of SFA has been an important aspect of human nutrition in recent years, and dietary alternatives that provide UFA are being sought. Ruminant meat does not fulfil these characteristics, as its UFA content is lower than that of pork and chicken meat (*Álvarez et al., 2020*). However, the inclusion of CT in the diet can modify the FA profile in the meat of small ruminants, and its effect is directly related to the diet, type, source and dose of tannins (*Vasta et al., 2019*; *Frutos et al., 2020*; *Pimentel et al., 2021*). In the present study, CT supplementation did not produce a consistent change in the FA profile beyond the decrease in stearic acid and the increase in CLA. The lack of a consistent response in the present study may be related to the amount of these metabolites that actually reach

the tissue to exert their effect. Therefore, it is possible that the observed changes in the stearic acid and CLA depend on the microbiota responsible for biohydrogenation in the rumen (*Frutos et al., 2020*). Diets supplemented with CT could positively interfere with UFA metabolism by reducing rumen biohydrogenation and increasing tissue deposition of these FA (*Min & Solaiman, 2018*).

Increased deposition of UFA, as observed in T4, improves the nutritional quality of meat (*Parodi, 2016*). The FA proportions observed in the present study are within the reference ranges observed in lamb meat under similar production conditions (*Manso et al., 2011*). In addition, SFA remained within the range of 43.1% to 44.2% in the evaluated treatments, with lower levels for T3 and T4, although the differences were not significant (Table 6). The monounsaturated FA (MUFA) content was between 47.4% and 48.5%, with oleic acid being the most abundant. The PUFA content was between 5.98% and 7.6%. In general, these SFA, MUFA and PUFA levels are in line with those reported by *Gómez et al. (2018)*.

The nutritional value of fat consumed in the human diet can be assessed by the PUFA-to-SFA ratio, which should be > 0.45 (*Alfaia et al., 2010*). *Gesteira et al. (2018)* showed that the inclusion of CT in the diet of lambs promoted a mean PUFA-to-SFA ratio of 0.35, reaching values close to the above recommendation. However, in the present study this ratio was below this value in the evaluated treatments (0.13 to 0.17). Hence, consumers would need to consume PUFA from other nutritional sources.

## CONCLUSION

The CT of AME showed high bioaccessibility, but only a small proportion was deposited in the meat. Supplementation of the lamb diet with CT from AME did not reduce lipid oxidation, nor did it affect SF or CWL, but it did improve some aspects of meat colour and CLA deposition, and there was evidence of a reduction in UFA when 0.5% or 0.75% AME was used. Due to the low levels of CT detected in the meat, most of the effect of these metabolites may be related to inhibition of rumen biohydrogenation.

## ACKNOWLEDGEMENTS

This work is a result of the LGAC (Line of Generation and Application of Knowledge): Efficient livestock farming, sustainable welfare and climate change, Colegio de Postgraduados interdisciplinary working group.

### Funding
The authors received no funding for this work.

### Competing Interests
The authors declare there are no competing interests.

## Author Contributions

- Alejandro García Salas conceived and designed the experiments, authored or reviewed drafts of the article, and approved the final draft.
- Jose Ricardo Bárcena-Gama conceived and designed the experiments, authored or reviewed drafts the article, and approved the final draft.
- Joel Ventura analyzed the data, prepared figures and/or tables, and approved the final draft.
- Canuto Muñoz-García analyzed the data, prepared figures and/or tables, and approved the final draft.
- José Carlos Escobar-España analyzed the data, prepared figures and/or tables, and approved the final draft.
- Maria Magdalena Crosby performed the experiments, authored or reviewed drafts of the article, and approved the final draft.
- David Hernandez conceived and designed the experiments, authored or reviewed drafts of the article, and approved the final draft.

## Animal Ethics

The following information was supplied relating to ethical approvals (i.e., approving body and any reference numbers):

The Animal Welfare Committee of the Colegio de Postgraduados fully approved this research (COBIAN/003/21).

## Data Availability

The raw measurements are available in the Supplementary File.

## Supplemental Information

Supplemental information for this article can be found online at http://dx.doi.org/10.7717/peerj.17572#supplemental-information.

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
