# Peer review of "Bioaccessibility of condensed tannins and their effect on the physico-chemical characteristics of lamb meat"

_PeerJ, doi:10.7717/peerj.17572_

## Round 0.1 · original submission · Major Revisions

Please make sure to respond to the comments made by the reviewers and address them in a rebuttal letter.

**Language Note:** PeerJ staff have identified that the English language needs to be improved. When you prepare your next revision, please either (i) have a colleague who is proficient in English and familiar with the subject matter review your manuscript, or (ii) contact a professional editing service to review your manuscript. PeerJ can provide language editing services - you can contact us at [email protected] for pricing (be sure to provide your manuscript number and title). – PeerJ Staff

Reviewer 1 ·

Basic reporting

no comment

Experimental design

no comment

Validity of the findings

no comment

Additional comments

The revised manuscript is interesting; however, it is necessary to address the following comments:

Most important comments

Line 30,31,34,35: If the meaning of the treatments was previously indicated, it is not necessary to indicate the percentages, only the assigned abbreviation
Line 32: Instead of assigning numerical values with the literal ones, it could indicate the order of bioaccessibility of the evaluated treatments from highest to lowest or vice versa.
Line 125: physico-chemical properties?
Line 141: What was the concentration in which trichloroacetic acid was used?
Note: add a first section at the beginning of the materials and methods section, describing the materials and reagents used, as well as the marks of origin
Line 144: What was the concentration was the TBA solution used?
Line 147: rewrite… mg of malondialdehyde (MDA) per kg of meat
Note: What was the reason why the pH parameter was not measured, if it is one of the most important physicochemical parameters to evaluate quality?
Line 192: This information must appear before the physicochemical properties, to respect the order of the title of the work and the objectives.
Line 222: (P<0.05) or (P < 0.05), homogenize terms through the manuscript
Line 227: use only the abbreviations for the color parameters, their meaning was previously described in the materials and methods section, review through the manuscript
Line 232: rewrite… T1 had….
Line 275: Previously the term PUFA was not used to abbreviate
Next most important comments

Line 22: What is the meaning of LW?
Line 23: If the abbreviation is not used later, remove (T) from text
Line 27: Insert storage temperature
Line 52: add examples of the used antioxidants
Line 57: after the word condensed tannins insert between parenthesis the abbreviature like in the abstract, and used it through the document
Line 59: after the word fatty acids insert between parenthesis the abbreviature like in the abstract, and used it through the document
Line 79: It is important to use abbreviations once they are previously mentioned in the text (CT, AME)
Line 101: What´s the meaning of MS?
Line 128: indicate the purpose of exposing the meat samples at room temperature for 30 min
Line 128: indicate the value of the room temperature
Line 142: insert equipment information, vortex
Line 142: What were the sample homogenization conditions, rpm, time, temperature?
Line 144: insert equipment information, water bath
Line 159-164: What units were used to express the results?
Line 174: the names of some reagents appear unabbreviated, and others abbreviated with their chemical formula, it is necessary to homogenize
Line 179,209: Which temperature was used during the centrifugation process?
Line 286: missed information at the end of line?
Line 287: use italic text format for scientific names
Line 298: The term was previously abbreviated in the materials and method section. It is not necessary to describe the term and its abbreviation again.

Least important points
Line 56: remove a dot after the word health
Line 121: Ziploc®, Ziploc® or Ziploc, homogenize terms through the manuscript
Line 122: shelf life or shelf-life, homogenize terms through the manuscript
Line 170: 0.5 M
Line 196: 24 h
Line 199: -20 °C

Reviewer 2 ·

Basic reporting

In this manuscript, Garcia-Salas et al. evaluate the bioaccessibility of condensed tannins and their effects on the physicochemical characteristics of fattening lamb meat. The paper is well-written, and the data were properly collected. However, the novelty of this work does not meet the standards of PeerJ. The study fails to deliver robust conclusions that could advance the field or contribute new insights. Unfortunately, I do not think the current form of this work is a good fit for the readers of PeerJ.

Experimental design

No comment

Validity of the findings

no comment

---

## Round 0.2 · Minor Revisions

There are small items to revise:

- the data representation/marking is confusing in the tables with e.g. "abcd Means with different letters in each row indicate significant differences (P ˂ 0.05)" - if all means the same, why the different letters; what each letter signifies should be clearer (this applies to most if not all tables).

- I think certain materials should be identified with respective catalog # and, in the case of vaccines, even potentially lot #.

- It is also confusing that the abstract starts directly with the objective, without an introductory sentence.

Please address these minor issues at your convenience.

Reviewer 1 ·

Basic reporting

No comments

Experimental design

No comments

Validity of the findings

No comments

Additional comments

The authors followed the recommendations made in the first review

Reviewer 2 ·

Basic reporting

The authors properly addressed my concerns, and I would recommend acceptance in its current form.

Experimental design

No comment

Validity of the findings

No comment

---

## Round 0.3 · accepted · Accept

Thanks for addressing the revisions requested. Now, your manuscript has been accepted by PeerJ.